# Covalent docking and molecular dynamics simulations reveal the specificity-shifting mutations Ala237Arg and Ala237Lys in TEM beta-lactamase

**Gabriel Monteiro da Silva**[1]*, **Jordan Yang**[2], **Bunlong Leang**[3], **Jessie Huang**[4], **Daniel M. Weinreich**[5], **Brenda M. Rubenstein**[2]

1 Department of Molecular and Cell Biology, Brown University, Providence, Rhode Island, United States of America, 2 Department of Chemistry, Brown University, Providence, Rhode Island, United States of America, 3 Department of Health and Human Biology, Brown University, Providence, Rhode Island, United States of America, 4 Department of Chemistry, Wellesley College, Wellesley, Massachusetts, United States of America, 5 Department of Ecology and Evolutionary Biology, Brown University, Providence, Rhode Island, United States of America

* gabrielmds@brown.edu

**Data Availability Statement:** The data that support the findings of this study are also openly available

## Abstract

The rate of modern drug discovery using experimental screening methods still lags behind the rate at which pathogens mutate, underscoring the need for fast and accurate predictive simulations of protein evolution. Multidrug-resistant bacteria evade our defenses by expressing a series of proteins, the most famous of which is the 29-kilodalton enzyme, TEM β-lactamase. Considering these challenges, we applied a covalent docking heuristic to measure the effects of all possible alanine 237 substitutions in TEM due to this codon's importance for catalysis and effects on the binding affinities of commercially-available β-lactam compounds. In addition to the usual mutations that reduce substrate binding due to steric hindrance, we identified two distinctive specificity-shifting TEM mutations, Ala237Arg and Ala237Lys, and their respective modes of action. Notably, we discovered and verified through minimum inhibitory concentration assays that, while these mutations and their bulkier side chains lead to steric clashes that curtail ampicillin binding, these same groups foster salt bridges with the negatively-charged side-chain of the cephalosporin cefixime, widely used in the clinic to treat multi-resistant bacterial infections. To measure the stability of these unexpected interactions, we used molecular dynamics simulations and found the binding modes to be stable despite the application of biasing forces. Finally, we found that both TEM mutants also bind strongly to other drugs containing negatively-charged R-groups, such as carumonam and ceftibuten. As with cefixime, this increased binding affinity stems from a salt bridge between the compounds' negative moieties and the positively-charged side chain of the arginine or lysine, suggesting a shared mechanism. In addition to reaffirming the power of using simulations as molecular microscopes, our results can guide the rational design of next-generation β-lactam antibiotics and bring the community closer to retaking the lead against the recurrent threat of multidrug-resistant pathogens.

in the project's GitHub Repository: https://github.com/GMdSilva/b_lactamase_results.

**Funding:** This work was funded by the Office of Experimental Program to Stimulate Competitive Research (EPSCoR, https://www.nsf.gov/od/oia/programs/epscor/) Track-II award number OIA1736253, awarded to BR and DW. The funders had no role in study design, data collection and analysis, decision to publish, or preparation of the manuscript.

**Competing interests:** The authors have declared that no competing interests exist.

## Author summary

Resistance to antibiotics is a major public health threat. Microorganisms are able to resist commonly used drugs by evolving and expressing enzymes capable of neutralizing antibiotics. Understanding the relationships between structural elements in these enzymes and their drug-clearing functions can lead to crucial insights for the discovery of next-generation antibiotics. In this study, we have used cutting-edge computational modeling methods to predict the effects of all naturally-occurring variations of an important region of the binding site of TEM $\beta$-lactamase, one of the major resistance-granting enzymes in bacteria. In an effort to identify patterns that could be useful for drug discovery, our simulations sought to understand how chemical changes in the tested region can affect resistance against a collection of over 90 widely used antibiotics. Crucially, through our simulations, we have identified a pathway for bacterial resistance against $\beta$-lactam antibiotics containing a negatively-charged moiety. We have also elucidated the mechanism behind the gain of resistance, which involves strong interactions between the drug's negative moieties and the positively-charged chemical shifts resulting from the mutation. Finally, we validated our predictions against fitness experiments for two commonly used antibiotics, which qualitatively corroborated our most important findings.

## Introduction

Predicting the fitness landscapes of pathogens is vitally important for medicine. The accurate mapping of the evolutionary paths of disease-causing agents is not only useful for broadening our understanding of biology and natural selection, but also has clear implications for public health. For instance, such predictions can turn the tide against the ever-increasing threat of drug-resistant pathogens that has arisen due to decades of the improper use of antibiotics [1]. Although billions of dollars are spent every year with the goal of developing ever-improved drugs that can eliminate resistant bacteria, endeavors exclusively focused on drugging the pathogens of today without an eye towards the pathogens of tomorrow are perilously near-sighted since pathogens evolve orders of magnitude faster than novel compounds that can be discovered and tested [2]. Predicting how pathogens evolve well before they do thus remains an outstanding and important challenge that has motivated thousands of studies around the world, leading to the explosive growth of the field of evolutionary forecasting, which seeks to use cutting-edge technologies enabled by the information revolution to model and predict the fitness landscapes of pathogens [3,4].

Traditionally, heuristics that seek to predict the effects of point mutations on the fitness of organisms rely on two principles: that the fitness of mutations is, firstly, inversely proportional to the degree of evolutionary conservation of the codon into which they are substituted, and secondly, the size of the chemical shift associated with the substitution of one residue for another [5–8]. The first principle rests on the often reasonable assumption that codons that are conserved across different species are likely important for functions, including folding, binding, and catalysis [9]. Thus, changing highly conserved positions often leads to negative fitness outcomes [6]. The chemical consequences of mutations are also important [8]. Substitutions of small polar amino-acids such as serine or threonine in the active sites of enzymes by the bulky aromatic residues tryptophan or tyrosine tend to be catastrophic, and mutations to proline are well-known to significantly affect folding [7]. Considering the above, it seems

reasonable to assume that mutations in conserved codons that lead to significant chemical shifts will be deleterious.

Although obviously useful for genome-wide predictions across thousands of species, these heuristics can be limited. Crucially, they do not account for the fact that proteins are highly flexible systems, often capable of accommodating theoretically high-impact mutations with no significant reduction in function. One classic example of this phenomenon is the Abelson kinase 1 Glu255Lys mutation, which is often predicted to be deleterious to function due to the glutamic acid to lysine charge inversion, but actually leads to pan resistance against inhibitors with minimal decreases in phosphorylation [10]. This mutant showcases how mutations resulting in drastic chemical shifts in conserved sites can even increase function under the right conditions. Precisely quantifying the sometimes subtle effects of mutations on binding energies is therefore pivotal for maximizing the accuracy of fitness predictions.

One suite of tools that are becoming increasingly more reliable for predicting fitness landscapes are computational techniques, including traditional [11–13] or statistical-learning-based molecular modeling [14,15] and binding affinity prediction tools, such as docking programs [16] and free-energy calculations [17–20]. These techniques make use of physics-based force fields [21] to predict the interactions between atoms in proteins and ligands to in turn predict how the structures, dynamics, and binding affinities of biomolecules will change upon mutation. While the accuracy of these methods is steadily improving, disagreements still arise between simulated predictions and experiments, as force fields and scoring functions are inherently compact approximations to the more complex, underlying physical chemistry and conformational sampling algorithms frequently undersample conformational space to improve efficiency [22].

To meet the challenge of balancing accuracy and throughput in the prediction of binding affinities, computational methods have been developed that assume an induced-fit mechanism, where ligands will provoke only slight conformational perturbations to the protein binding site upon interactions with the receptor. In particular, covalent docking methods have been developed specifically to predict the binding affinities of compounds that bind covalently (either permanently or as reaction intermediates) to receptors. Given the relatively high accuracy of covalent docking for predicting substrate binding, we sought to predict the effects of thousands of point mutations on the binding affinity of TEM $\beta$-lactamase to common classes of $\beta$-lactam antibiotics, with an eye toward mutations whose fitness outcomes elude common heuristics.

As its name suggests, TEM-1 is capable of hydrolyzing the $\beta$-lactam group of penicillin and other $\beta$-lactam-ring-containing drugs, rendering them inactive and allowing bacteria to continue to thrive [23], which represents a grave public health threat. TEM's enzymatic mechanism has been extensively documented [24,25]. After the binding of a $\beta$-lactam compound to its active site, the side-chain oxygen of TEM's Ser70 residue binds to the $\beta$-lactam's carbonyl via nucleophilic attack, opening the $\beta$-lactam ring and forming a transient covalent bond between the enzyme and the drug [25]. Finally, the bond is broken in the deacylation step during which Ser70's hydroxyl is reconstituted and the inactivated antibiotic is released [25] (S1 Fig). This intricate mechanism requires the aforementioned formation of hydrogen interactions between the oxygen of the $\beta$-lactam's carbonyl and the backbone amide hydrogens of residues Ala237 and Ser70, which draw electrons away from the carbonyl, enabling the nucleophilic attack. This structure is known as an oxyanion hole and is vital for substrate recognition and cleavage [25].

Currently, over 90 $\beta$-lactam antibiotics are available for the treatment of bacterial infections, many of which are regularly prescribed in high-risk clinical environments. Being able to predict how this wide array of drugs will interact with the numerous possible $\beta$-lactamase mutants

requires a robust array of relatively high-throughput predictive strategies. Historically, researchers have achieved modest amounts of success using molecular docking to predict the binding affinities of $\beta$-lactam antibiotics against a series of clinically-prevalent TEM mutants (with 3D structures obtained via crystallography or homology modeling [26]). Previous predictions performed using static structural snapshots have been vastly improved by incorporating an ensemble of sequential conformations obtained via building Markov chains from multiple-microsecond molecular dynamics (MD) simulations [26]. This gain in predictive accuracy is rooted in the fact that TEM's active site is relatively occluded and shifts considerably to accommodate the binding of bulkier substrates, which is a phenomenon that requires MD or induced-fit/covalent docking to properly capture. Although extremely useful for focused studies, the computational cost associated with obtaining the long trajectories required for building the Markov chains is prohibitive for high-throughput projects that seek to measure the effect of dozens of individual point mutations on the binding of hundreds of different compounds, highlighting the necessity of alternative ways for studying the details of TEM's active site without sacrificing celerity.

Rising to this challenge, we predicted the effects of 19 Ala237 mutations on the binding of 91 $\beta$-lactam antibiotics to $\beta$-lactamase. In particular, we identified two previously-unreported, specificity-shifting mutations (Ala237Arg/Lys) that counter-intuitively increase the binding affinity of certain $\beta$-lactam drugs despite the fact that these mutations involve the substitution of a smaller, hydrophobic residue with bulkier, charged residues that would normally be expected to reduce binding. To balance computational efficiency with sufficient conformational sampling of both the protein and the ligands, we initially identified these mutations through covalent docking and computationally affirmed our predictions using molecular dynamics simulations. We further validated these predictions by measuring the experimental resistance of *E. coli* containing Ala237 point mutations treated with ten of the drugs studied and compared this resistance data with our predictions. Positive correlations were found between our predictions and the experimental results, with the strongest correlations observed for the penicillin, ampicillin, and the cephalosporin, cefixime. Further interrogation through molecular dynamics simulations, revealed that Ala237Arg leads to active site steric clashes with ampicillin, but surprisingly enhances the binding of the larger cefixime. We observed that Ala237Lys undergoes nearly identical specificity shifts, and identified a specific R-group in cefixime, also present in other $\beta$-lactam antibiotics such as carumonam and ceftibuten, that foster these unexpected interactions.

The change from the small, hydrophobic alanine to bulky, positively-charged residues represents a significant chemical shift, but one that is nevertheless accommodated by the TEM active site. The unexpected increases in binding we observed for mutations of the catalytic Ala237 to arginine and lysine are thus a testament to the importance of accounting for receptor flexibility when predicting binding affinities. Ultimately, our findings may help guide next-generation strategies for the rational design of $\beta$-lactam antibiotics, and reveal valuable information about enzymatic mutational landscapes and biochemistry.

## Results and discussion

### Active site mutants and clustering by drug class

In order to study shifts in binding affinity, we began our analyses by obtaining the docking scores of 91 $\beta$-lactam ligands in complex with 19 mutant TEM structures using CovDock (see the S1 Appendix for a more detailed discussion of this choice) and then dividing these scores by the docking scores of the same drug in complex with the wild-type TEM-1 structure. CovDock is a covalent docking algorithm developed by Schr¨odinger, Inc., that uses the Glide

**Fig 1. Schr¨odinger's CovDock workflow for covalent docking, adapted from Schr¨odinger's documentation (release 2020–1).**

docking engine [16] coupled to the Prime molecular modeling software to covalently dock compounds. As summarized in Fig 1, the CovDock workflow starts by mutating the reactive residue (in this case, Serine 70) to alanine, then enumerates ligand conformations, docks the ligand to a flexible receptor with positional restraints (to ensure the formation of the covalent bond), restores the reactive residue to a rotamer predicted to be favorable, and forms the covalent bond. Finally, the energy of the binding site is minimized and docking scores for the covalent poses are calculated with Glide. This process yielded a relative docking score for each ligand and mutant structure, which was used in all subsequent analyses (Fig 2). Since many mutations are similar in outcome and lead to negligible changes in the predicted binding affinity for most compounds, we expect that docking score differentials of at least 25% in either direction represent a significant change in predicted affinity. The most pronounced differences between the wild-type and mutant docking scores were observed when the relatively small alanine residue was substituted with large aromatic residues, presumably because they result in active site occlusion leading to steric clashes. In many cases, these substitutions caused CovDock to fail to find any favorable poses, as indicated in black in Fig 2, even using the extended sampling protocol which generates a larger set of poses for each ligand. Conversely, the substitution of alanine for the chemically-similar amino acids valine or serine did not affect binding considerably. Besides these two clear patterns, we also identified a few surprising trends, such as the Ala237Lys mutation improving the binding of most cephalosporins—unexpected given the bulkiness of lysine.

To better understand the molecular basis behind the shifts in docking scores observed for each mutant, we classified compounds based on their respective drug classes and repeated our previous analysis. This was done under the assumption that chemically-alike compounds would be affected in similar ways by a given change to the active site of the enzyme, and that

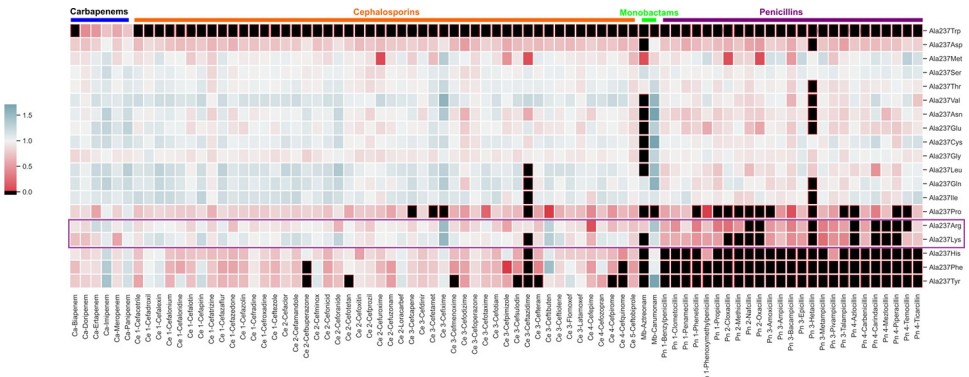

**Fig 2. Clustered heatmaps of relative docking scores for each docked compound against TEM β-lactamase structures harboring mutations in active site residue alanine 237.** Black indicates that the docking algorithm could not find poses that bind in realistic conformations to the target due to overwhelmingly unfavorable interactions that lead to nonphysical energies, such as steric clashes introduced by the aromatic substitutions. Docking scores are normalized with respect to the wild type docking score of each compound.

understanding the mechanisms behind these changes would pave the way for the use of predictive technologies and rational drug design.

We classified our drugs into four main groups: penicillins, cephalosporins, monobactams, and carbapenems. Penicillins and cephalosporins were also subdivided based on generation: first-, second-, third- (aminopenicillins), and fourth-generation for penicillins; and first-through fifth-generation for cephalosporins. We then grouped results by relative docking score using hierarchical clustering [27]. Mutations to Ala237 led to a revealing clustering by drug class (Fig 3). The binding of most penicillins was significantly decreased by the substitution of an alanine with a larger residue such as an arginine or lysine, whereas the binding of cephalosporins was not as strongly affected.

Considering the above, we decided to probe which compound/mutant TEM pairs led to interesting and unexpected specificity-shift phenotypes. By contrasting the predictions obtained from CovDock, we identified that the Ala237Arg and Ala237Lys mutations seem to reduce the binding of penicillins across the board, while increasing TEM's affinity for oxyimino-cephalosporins such as cefixime or ceftazidime by up to 30%. This result is particularly interesting because both arginine and lysine are positively-charged and relatively bulky in comparison to alanine, thus significantly changing the topology and electrostatic profile of the binding site (S2 Fig). Among the cephalosporins whose binding affinity to TEM is predicted to be enhanced by this class of alanine substitutions, the binding of the third-generation oxyimino-cephalosporin cefixime displays the most dramatic change. In the Ala237Arg mutant, cefixime's CovDock Score increased by upwards of 50% in relation to its predicted binding affinity against wild-type TEM, and the Ala237Lys change leads to a similar outcome (S3 Table). Interestingly, cefixime carries an extended carboxylic acid R-group that is often negatively-charged at physiological conditions, which could explain its enhanced recognition and stabilization by positively-charged residues in the binding site. Before testing this hypothesis using more rigorous simulation methods, we sought to validate these results with experimental inhibition assays in order to confirm that these changes are in fact real and not artifacts of binding score functions. Finally, fifth-generation cephalosphorins (such as ceftobiprole) tend

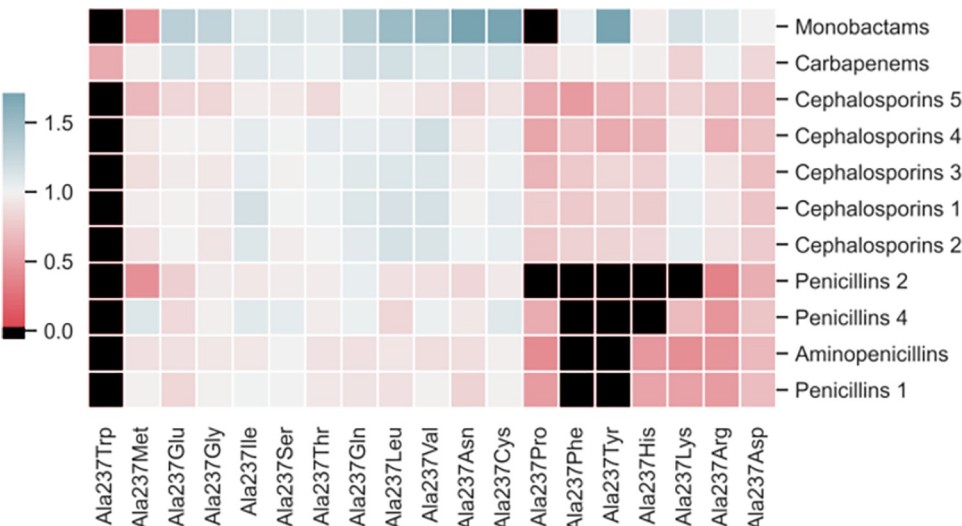

**Fig 3. Clustered heatmaps of relative docking scores for each drug class against TEM $\beta$-lactamase structures harboring mutations in active site residue alanine 237.** Black indicates that no favorable poses were found, suggesting that the ligand was unable to bind. Docking scores are normalized with respect to the wild-type docking score.

to be considerably bulkier than their earlier counterparts, which provides a plausible hypothesis as to why the substitutions to arginine or lysine were not capable of improving their binding affinities against TEM. Alanine 237 is particularly important for substrate recognition and catalysis as its backbone, along with that of Serine 70, forms the oxyanion hole that draws electrons away from $\beta$-lactams' carbonyl groups [25], thus enabling acylation by TEM. Many of the bulky substitutions to Ala237, such as Ala237Arg, weaken this important interaction, which is reflected in their reduced binding affinities. Importantly, Alanine 237 is conserved amongst species that express TEM, with a PRCP conservation score [28] of 0.6 calculated from the 50 entries most similar to *E. Coli's* TEM-1 in the UniProt database (UniProt ID P62593) [29].

## Comparison with experiment

As described in the Methods section, we used minimum inhibitory concentration (MIC) assays to test the antibiotic resistance of *E. coli* expressing 20 different TEM constructs, spanning all possible substitutions of TEM's codon 237, upon treatment with varying concentrations of either cefixime or ampicillin. These two compounds were selected as representative of their drug classes (third-generation aminopenicillins and cephalosporins, respectively) due to promising results obtained in the prediction part of this study. To obtain a measure of gain or loss of fitness in response to each substitution, we divided the MIC of bacteria expressing each mutant upon treatment with each drug by the MIC of bacteria expressing wild-type TEM upon treatment with the same drug. Finally, we assessed the agreement between prediction and experiment (Fig 4).

We observe that both the Ala237Arg and Ala237Lys substitutions led to substantial increases in the resistance of the *E. coli* mutant constructs against cefixime, corroborating the prediction results and strengthening the hypothesis that the charged substitutions have stabilizing effects for cefixime, in lieu of their ability to bind ampicillin. Interestingly, Ala237Thr also leads to an equivalent increase in resistance, in contrast with CovDock's results, which predict its binding affinity to be slightly decreased compared to that of the wild-type. This

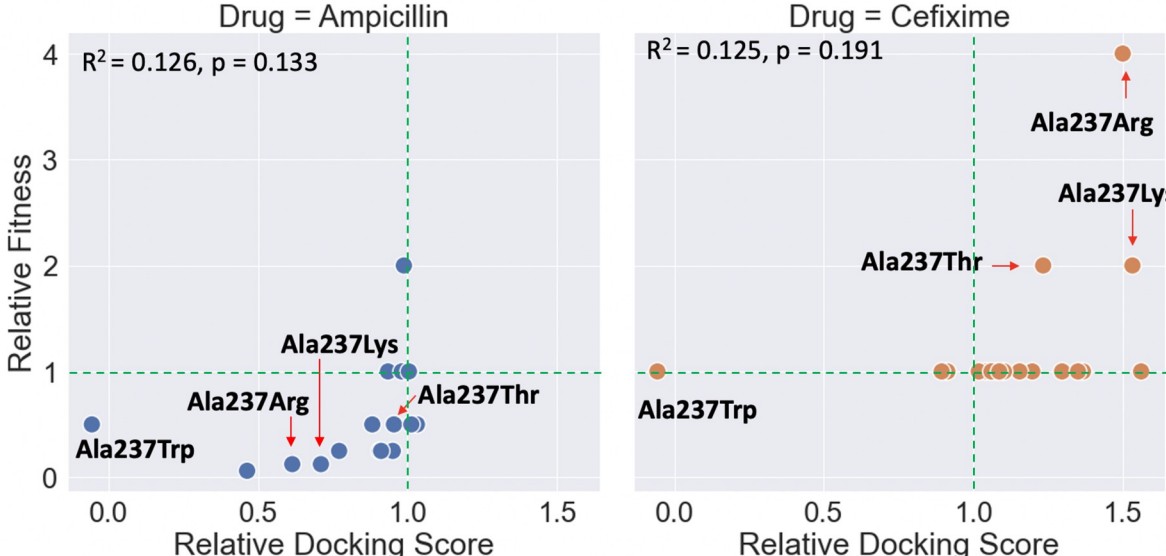

**Fig 4. Comparison between changes in fitness upon treatment with either ampicillin (left) or cefixime (right) for *E. coli* carrying mutations in TEM's codon 237 and the relative docking scores of each compound bound to each mutant structure.** Results are tabulated in S4 Table.

suggests that the Ala237Thr substitution might affect binding in complex ways, such as altering active site dynamics. Indeed, previous literature indicates that the Ala237Thr substitution is known to lead to enhanced-spectrum/cephalosporinase phenotypes, through a variety of proposed mechanisms [30,31]. Also of note, bacteria expressing TEM-Ala237Trp are highly susceptible to ampicillin and cefixime, which is corroborated by the modeling results. *In silico*, the Ala237Trp substitution markedly reduced the predicted binding of both drugs due to active site occlusion leading to steric clashes that drastically increased the energy barrier for the acylation reaction.

The main caveat of comparing relative docking scores to changes in minimum inhibitory concentrations is that many complex factors contribute to the ultimate fitness of an organism, not just a single compound's ability to bind to an enzyme [3]. Thus, while a drug's docking score to $\beta$-lactamase should correlate with *E. coli* susceptibility to that drug, these relationships may sometimes be obscured by other factors *in vivo*. Another important consideration when comparing predicted binding affinity to fitness is readout resolution. In theory, scoring functions are designed to identify and report minute changes in binding site topology, whereas MIC assays are discrete in nature and thus might not give a quantifiable readout between alternative conditions that further decrease an already minimal fitness for the specific treatment regimen being tested. In practice, this means that the results of MIC assays will often cluster around the baseline value if that baseline is already minimal (for example, in the case of bacteria that are susceptible to the drug being tested, such as cefixime), whereas predicted binding values will exhibit a much less skewed distribution, occupying values above and below the baseline since the slightest of changes will be detected by most scoring functions.

This phenomenon is observed when comparing the outcomes of the fitness after performing cefixime treatment assays and the cefixime binding affinities as predicted by CovDock. Since *E. coli* expressing wild-type TEM is susceptible to cefixime, the relative fitness results cluster around the baseline values as the vast majority of Ala237 mutations do not lead to enhanced binding of the drug. Conversely, wild-type TEM is very efficient at cleaving ampicillin, having been optimized by selection for that task. Thus, most Ala237 mutations have a negative impact on fitness.

A final caveat is that CovDock ranks poses through scoring functions that approximate the underlying electronic structure of the docking problem, which is useful for this study due to the large number of drugs and targets being tested, but comes with sacrifices in accuracy. Combined, these considerations provide reasonable explanations for the modest agreement between predicted relative binding affinity and experimental relative fitness.

Although the overall correlation is very weak, not allowing for broad hypotheses spanning all possible mutants, we identified very interesting patterns for a few substitutions. Considering the encouraging experimental results obtained for the Ala237Arg/Lys substitutions in the cefixime treatment, we sought to further test the proposed model for enhanced binding of the drug due to the formation of ionic interactions.

## Mechanistic analysis

Seeking to identify the molecular mechanism behind the potential specificity shift described above, we used Schr¨odinger's Maestro (Schr¨odinger Release 2020–1) [32] to visualize the complexes generated by docking and quantified favorable and unfavorable interactions between ampicillin/cefixime and TEM Ala237Arg. We then compared these interactions with those present in complexes with wild-type TEM.

Visual inspection reveals the most likely culprit behind the change in specificity from ampicillin to cefixime: the bulky side chain of arginine 237 occludes the mutant TEM active site,

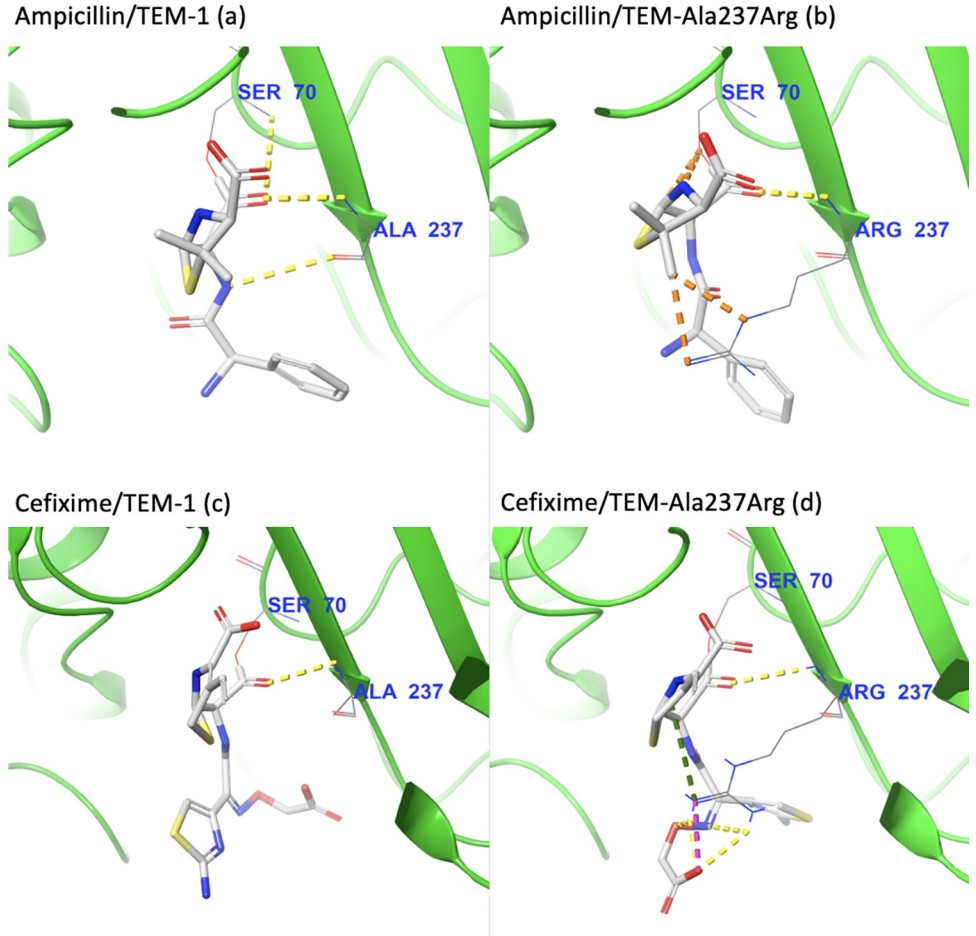

**Fig 5. Visual inspection of the intermolecular interactions with key active site residue 237 for ampicillin covalently bound to (a) TEM-1 and (b) TEM-Ala237Arg, and of cefixime covalently bound to (c) TEM-1 and (d) TEM-Ala237Arg.** Yellow dotted lines indicate hydrogen interactions, orange dotted lines indicate poor contacts/clashes, purple dotted lines indicate salt bridges, and green dotted lines indicate cation-pi interactions.

creating unfavorable interactions with ampicillin, preventing it from binding optimally and thus leading to a reduced docking score (Fig 5).

The opposite happens with cefixime: the positively-charged arginine strongly stabilizes the negative R-group of the drug, more than doubling the number of favorable interactions in comparison with the complex with wild-type TEM (Fig 5). Importantly, the interaction of the β-lactam carbonyl group with the backbone of Arg237 is still present, indicating that this pose probably permits catalysis.

These contrasting results are particularly intriguing because they go against the traditional wisdom that mutations that create steric clashes always adversely affect binding—as even the relatively large cefixime was able to bind strongly due to the changes in the electrostatic profile caused by the Ala237Arg mutation. In this specific case, it seems that the gain of a charged interaction more than makes up for the additional energy barriers presented by the bulky arginine substitution.

To test the persistence of the interactions described above and the overall stability of the complexes generated by CovDock, we conducted ten independent replicates of binding pose

metadynamics (BPM) simulations of each ligand/target complex. Specifically, for this test, we used the highest-scoring ligand pose prior to the formation of the covalent bond in order to allow free movement of the ligand with respect to the receptor. The simulation time was set to ten nanoseconds for each round, and the collective variable (CV) was defined as the root mean square deviation (RMSD) of the ligand with respect to its initial conformation and coordinates generated by Glide.

Trajectory analysis confirms that the ampicillin/TEM-1 complex is considerably more stable than that of cefixime/TEM-1, as ampicillin converged quickly and remained stable throughout the full ten nanoseconds simulations. Alternatively, cefixime showed a large degree of motion, shifting more than 2.5 A from the pose calculated by Glide (Fig 6). In accordance with previous results, the opposite is observed for the complexes with TEM Ala237Arg: cefixime converges after the two nanoseconds mark, while ampicillin's RMSD increases up until eight nanoseconds. This difference in motion is also reflected in the PersScore, which is considerably higher for the cefixime complex, indicating that cefixime is capable of maintaining more of its initial hydrogen interaction network. Of note, cefixime has twice the number of rotatable bonds as ampicillin (eight vs. four, respectively), which translates into a higher baseline RMSD.

As a further measurement of the stability of the complexes generated by CovDock, we quantified the number of intermolecular interactions between atoms of the ligands and of the active site residues (Ala237, Ser235, Arg244, Tyr105, Ser130, Ser70, etc.) for each complex over the course of one 10-nanosecond metadynamics simulation (Fig 7). This revealed a considerable shift in the interaction number, as ampicillin lost most of its hydrogen interactions in response to the Ala237Arg mutation, whereas cefixime more than doubled its hydrogen interactions and salt bridges with active site residues.

Finally, we also measured the interactions between each ligand and Arg237 for the TEM-Ala237Arg complexes over the time course of the simulations in order to isolate the specific contribution of the change from alanine to arginine in terms of the number of intermolecular interactions (Fig 8). As expected, Arg237 makes a significantly larger number of favorable interactions with cefixime than it does with ampicillin, including a salt bridge that is present through the entire ten nanoseconds of simulations, even under the effects of the bias potential. This salt bridge is formed by the positive moiety of the arginine and the previously discussed negatively-charged R-group of cefixime, which is absent in ampicillin.

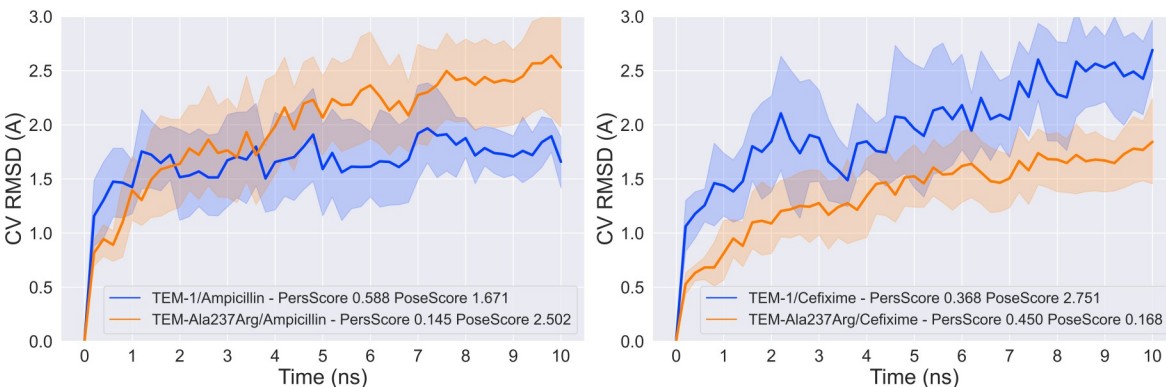

**Fig 6. Binding pose metadynamics results for complexes with ampicillin (left) and cefixime (right).** In addition to the ligand RMSD, for each set of simulations, we also report a measurement of the stability of the network of hydrogen bonds in the pre-simulation pose, called the PersScore, and the ligand RMSD for the final frame of the simulation in regards to the initial pose, called the PoseScore. All data shown are averaged over ten replicates. The shading denotes the standard deviation of the RMSD values for each system.

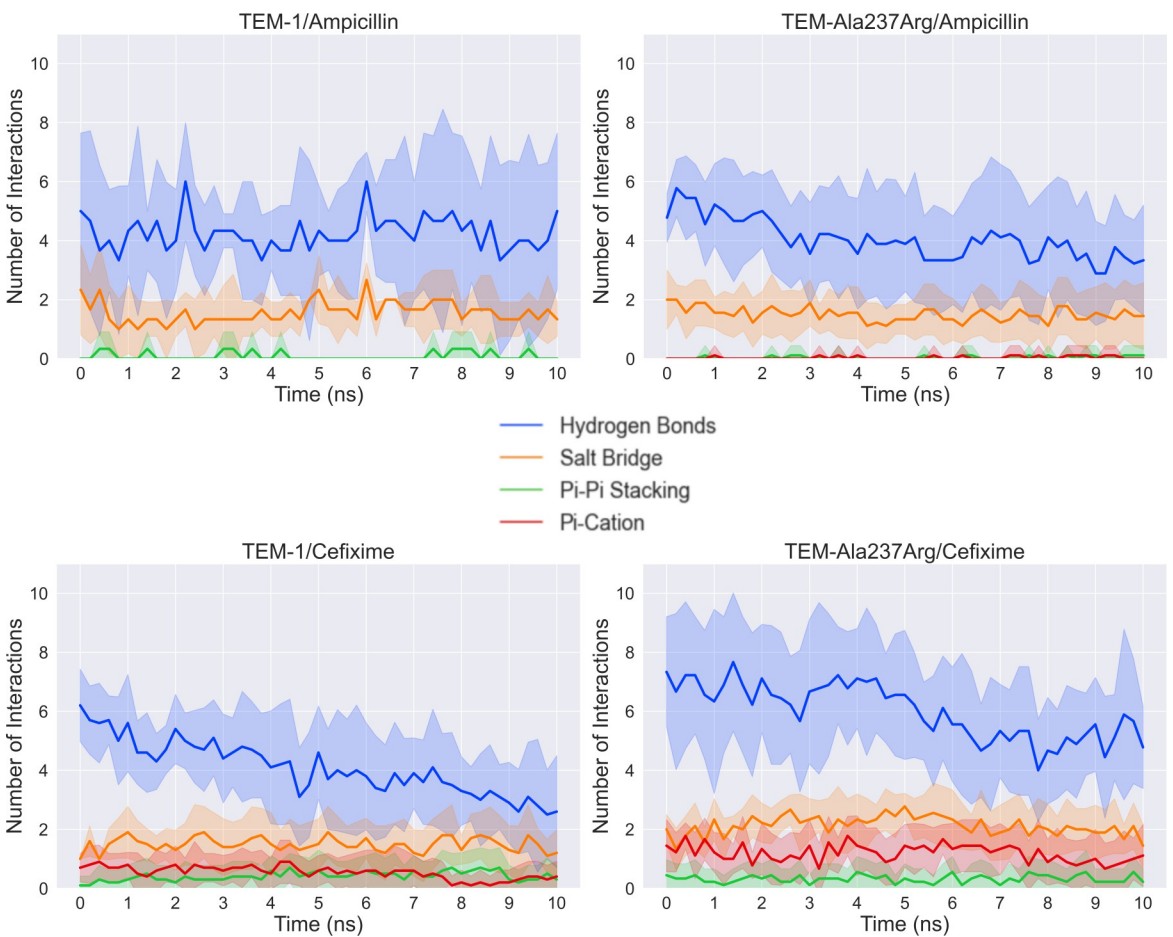

**Fig 7. Interaction analysis between the ligand and the active site residues of complexes with TEM-1 (left) and TEM-Ala237Arg (right).** Data are averaged from ten replicates of MD simulations. Ampicillin possesses a smaller number of pi interactions than cefixime because cefixime is more aromatic. In addition, pi interactions must fall into a very small range of angles to be considered present, and thus are more likely to be absent when the ligand binding mode is unstable. The shading denotes the standard deviation of the number of interactions for each system.

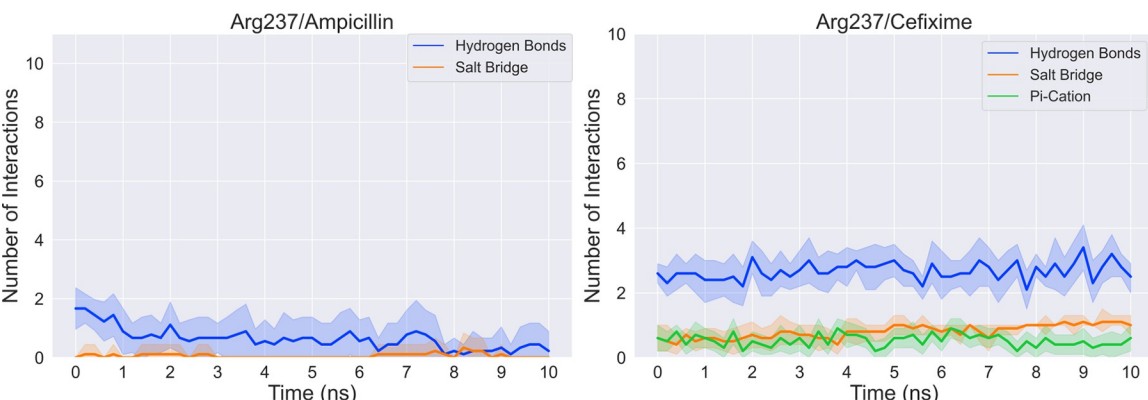

**Fig 8. Analysis of the number of interactions between ampicillin or cefixime and Arg237 of TEM-Ala237Arg complexes.** Data are averaged from ten replicates of MD simulations. The shading denotes the standard deviation of the number of interactions for each system.

### Generalization and insights for drug design

Having found a potential mechanism for changes in the substrate specificity of TEM $\beta$-lactamase mutants, we revisited our docking results to look for drugs that are structurally similar to cefixime. If our hypothesis—that the alanine to arginine substitution can counter-intuitively foster binding—is true, then we expect to see similar patterns for the binding of these drugs against TEM Ala237Arg.

Using Schrödinger Canvas (Schrödinger Release 2020–1), we ran leader-follower clustering using cefixime as a leader and circular extended connectivity molecular fingerprints [33] as descriptors. This workflow identified two compounds that are significantly similar to cefixime: ceftibuten (another cephalosporin) and carumonam (a monobactam). A comparison of the structures reveals that ceftibuten shares the same negatively-charged R-group as cefixime, and carumonam has a slightly shorter variation of it (Fig 9). A scaffold comparison is also reported in S3 Fig.

Since this R-group is stabilized by arginine 237 in the cefixime/TEM Ala237Arg complex, we asked if arginine 237 also interacts similarly with the R-groups of ceftibuten and carumonam. Indeed, visual inspection reveals that arginine 237 stabilizes ceftibuten and carumonam in similar fashions to the way it stabilizes cefixime: the positively-charged side-chain forms a salt bridge with the negatively-charged R-groups of each compound (Fig 10). These additional interactions are also reflected in the relative docking scores, which significantly increase for the docking of both ceftibuten and carumonam against Ala237Arg (121% and 138% of the docking scores for wild-type TEM, respectively).

Next, we asked if this effect was exclusive to the Ala237Arg mutation, or if changing alanine to another non-aromatic positively-charged residue would yield similar results. We focused on non-aromatic residues as changing alanine 237 to aromatic amino acids such as tryptophan, tyrosine, or phenylalanine seems to dampen predicted binding affinities across the board, likely due to steric clashes (Fig 3). Interestingly, an Ala237Lys mutation led to proportional shifts in the substrate specificity, as the predicted binding of penicillin severely decreased and the predicted binding affinities of cefixime and ceftibuten significantly improved.

For the Ala237Lys mutant, the positive moiety of lysine forms a salt bridge with the negatively-charged oxygen of each drug's carboxylic acid, equivalent to the interaction observed in complexes with the Ala237Arg mutant. The binding of ampicillin against this mutant is impacted by steric clashes between ampicillin's phenyl group and the side-chain of the mutated residue, leading to sharp decreases in predicted affinity (Fig 11).

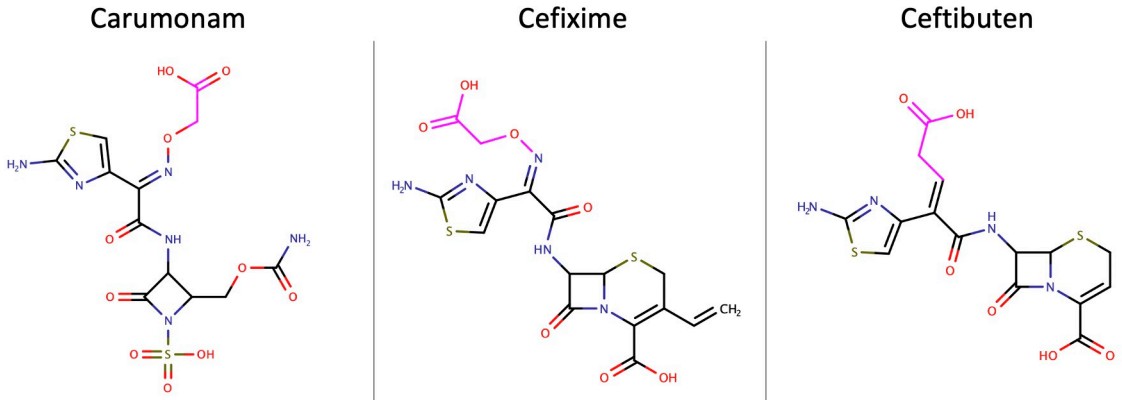

**Fig 9. Aligned two-dimensional structural representations of the cephalosporins cefixime and ceftibuten, and of the monobactam carumonam.** The negatively charged R-groups that are stabilized by the Arg237 mutation are highlighted in fuchsia.

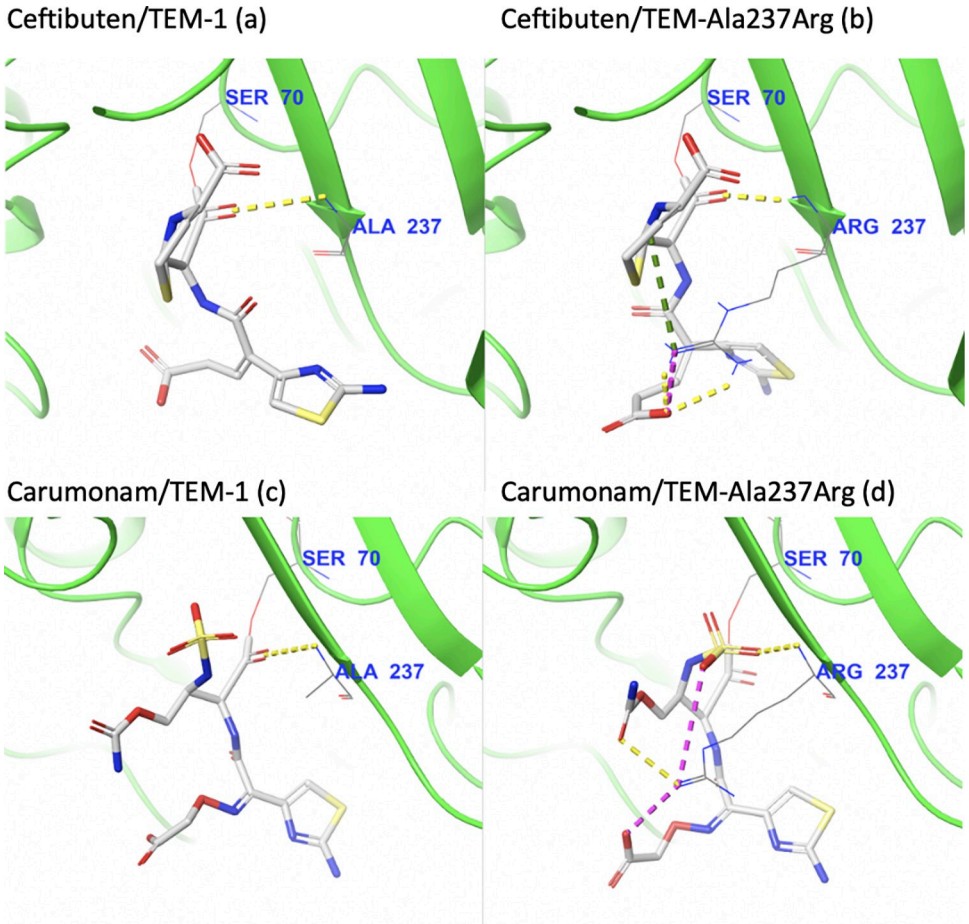

**Fig 10. Visual inspection of the intermolecular interactions with key active site residue 237 for ceftibuten covalently bound to (a) TEM-1 and (b) TEM-Ala237Arg, and for carumonam covalently-bound to (c) TEM-1 and (d) TEM-Ala237Arg.** Yellow dotted lines indicate hydrogen interactions, orange dotted lines indicate poor contacts/clashes, purple dotted lines indicate salt bridges, and green dotted lines indicate cation-pi interactions.

Finally, we also compared our results to the previously described experimental fitness dataset, and found qualitative agreements for the ampicillin and cefixime treatments. Unfortunately, our dataset lacks drug resistance values for treatments with carumonam and ceftibuten (Table 1).

## Conclusions

In this work, we employ a combination of high-throughput covalent docking followed by accelerated molecular dynamics simulations to analyze the effects of single mutations on the binding affinities of $\beta$-lactam drugs to TEM $\beta$-lactamase. Counterintuitively, our covalent docking studies reveal that mutating Ala237, which assumes a key role in $\beta$-lactam catalysis, to comparatively bulkier arginine or lysine residues results in larger binding affinities for certain cephalosporins (cefixime and ceftibuten) and carumonam. These docking results are corroborated by molecular dynamics simulations that demonstrate that the mechanism underlying the increased binding affinities stems from the formation of additional hydrogen bonds and salt bridges between the charged lysine and arginine residues and similar R-groups of the more strongly-bound compounds. Importantly, our computational findings are confirmed by

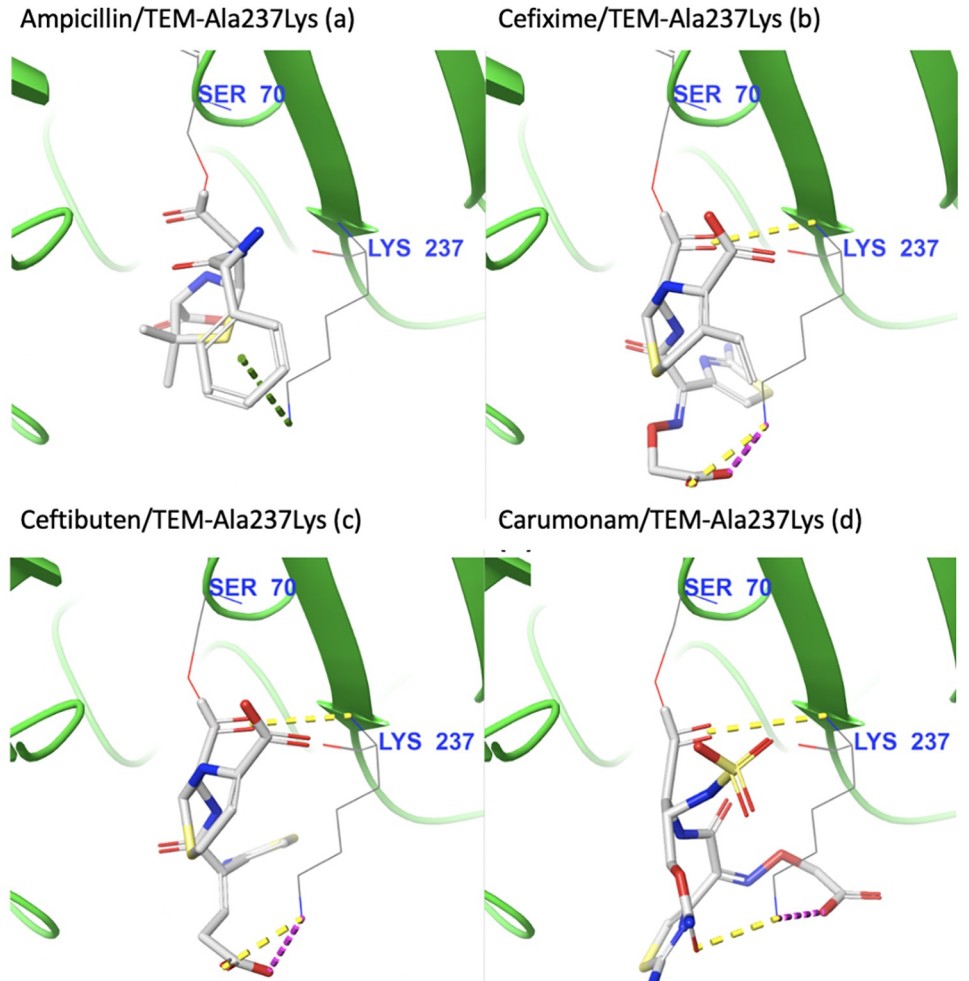

**Fig 11. Visual inspection of the intermolecular interactions with Lys237 for ligands covalently bound to TEM-Ala237Lys. (a) Ampicillin, (b) Cefixime, (c) Ceftibuten, and (d) Carumonam.** Yellow dotted lines indicate hydrogen interactions, orange dotted lines indicate poor contacts/clashes, purple dotted lines indicate salt bridges, and green dotted lines indicate cation-pi interactions.

minimum-inhibitory concentration assays that indicate that *E. coli* expressing TEM with the Ala237Arg and Ala237Lys mutations are more fit than the wild-type upon treatment with cefixime.

Excitingly, our results reveal two phenomena that go against traditional wisdom in the field: 1) that mutations in conserved residues that are involved in catalysis can lead to increased fitness against certain antibiotics; and 2) that mutations that lead to larger steric clashes with substrates entering the active site can paradoxically enable the binding of even bulkier substrates through active site rearrangements leading to favorable contacts and charged interactions with R-groups of the aforementioned drugs. Crucially, we have

**Table 1. Relative docking scores (RDS) and relative fitness values (when applicable) for TEM-Ala237Arg/Lys in complex with ampicillin (A), cefixime (C), ceftibuten (Cet), and carumonam (Car).**

| Mut. | A. Fit. | C. Fit. | A. RDS | Cei. | RDS | Cet. | RDS | Car. RDS |
|------|---------|---------|--------|------|-----|------|-----|----------|
| Ala237Lys | 0.125 | 0.1 | 0.71 | 1.55 | | 1.34 | | 1.17 |
| Ala237Arg | 0.125 | 0.3 | 0.614 | 1.52 | | 1.29 | | 1.41 |

demonstrated that these interactions are stable even under biasing forces in metadynamics MD simulations, and are a product of the substitutions from the neutral alanine to positively-charged non-aromatic residues, as other bulky mutations such as Ala237Tyr or Ala237Trp unsurprisingly reduce binding across the board. In brief, the unexpected favorable binding of cefixime and similar compounds against TEM Ala237Arg or Ala237Lys through induced-fit mechanisms demonstrates the importance of accounting for conformational dynamics when predicting ligand affinity.

Finally, we reiterate that the covalent docking method is a promising approach for measuring the effect of point mutations on the binding of compounds against the relatively occluded binding site of TEM $\beta$-lactamase, when other methods might fail to find favorable poses due to poor side-chain sampling. Although the correlation of docking results with experimental results is modest, the covalent docking method allows for the rapid testing of thousands of receptor conformations and compounds, thus drastically increasing the chances for detecting interesting binding phenomena.

We hope that this new understanding will be integrated into rational drug design strategies in order to minimize the ever-looming threat of antibiotic resistance, in addition to illustrating a clear case of an unexpected gain of fitness from mutations predicted to be deleterious by traditional methods.

## Materials and methods

### Conservation analysis of Alanine 237

To measure the conservation level of Ala237 across different species expressing TEM $\beta$-lactamase, we queried the UniProt database [29] for proteins with sequences similar to *E. Coli's* TEM-1 sequence (UniProt ID P62593). We then generated a multiple sequence alignment of the 50 most similar entries using the CLUSTAL W algorithm [34], and used the Protein Residue Conservation Prediction webserver [28] to calculate the conservation score of Alanine 237 based on the previously generated alignment.

### Homology modeling of Ala237 mutants

Using TEM-1 (PDB ID 1BTL) as a template, we generated all 19 non-synonymous standard amino acid substitutions of codon position 237 using Schrödinger's Prime side-chain prediction module Schrödinger Release 2020–1: Prime, Schrödinger, LLC, New York, NY, 2020). Side-chain conformations were scored in terms of their favorable and unfavorable interactions with neighboring atoms using the OPLS3e force-field [35], and the structure containing the lowest-energy configuration of each mutant post steepest-descent energy minimization was chosen for downstream analysis.

### Ligand selection

A set of 91 commercially-available $\beta$-lactam antibiotics of known activity were selected from PubChem (S2 Table) [36]. Three-dimensional Mol2 structure files for each ligand were downloaded from the ZINC database [37] and physiologically-relevant protonation states for each ligand were generated using LigPrep (Schrödinger Release 2020–1: LigPrep, Schrödinger, LLC, New York, NY, 2020).

### Molecular docking

Ligands were covalently docked against different TEM $\beta$-lactamase mutants via a $\beta$-lactam ring-opening reaction using Schrödinger's CovDock (Schrödinger Release 2020–1:

CovDock, Schr̈odinger, LLC, New York, NY, 2020) [35]. CovDock enables covalent docking of small molecules to proteins by first mutating the reactive residue to alanine, docking the ligand to the active site using extensive conformational sampling with certain translational constraints (to prevent the reactive payload from moving away from the binding residue), and then mutating the reactive residue back to its original state and building the covalent bond to the ligand's payload. The complex-bound ligand is then minimized and binding energies are predicted using a scoring function that takes into account both the affinity of the initial docking and the conformational strain associated with the covalent binding and subsequent active site rearrangements (ligands that require significant structural shifts to accommodate receive less favorable scores). Serine 70 was defined as the reactive residue in TEM-1, based upon significant evidence from the literature [38]. The thoroughness was set to "pose prediction" for increased accuracy. For a clearer visualization of the CovDock heuristic, see Fig 1.

## Molecular dynamics simulations

Binding pose metadynamics of the ligand/receptor complexes prior to the formation of the covalent bond were conducted using Schr̈odinger's Desmond with the OPLS3e force-field [39]. Metadynamics is a sampling technique with a plethora of applications in chemistry and biology, but most commonly employed to reconstruct free energy landscapes [40]. For our analyses, the choice to use binding pose metadynamics is justified on the basis that it allows us to measure ligand movement and ligand/active site residue interactions as a function of time under biases (that are ultimately removed). In our simulations, we set the collective variable to be the root-mean-square deviation (RMSD) of the heavy ligand atoms' positions relative to the positions of the same atoms in the initial (docked) pose previous to the formation of the covalent bond, with the aim of encouraging ligand fluctuations that can more efficiently explore conformation space in a short time frame. This decision was rooted in the intuition that ligands that interact strongly with active site residues will be less affected by the biases over the time course of the simulations, as their favorable intermolecular interactions provide energy barriers that must be surmounted before the ligand is able to move freely. Conversely, ligands that bind poorly will more readily succumb to the biasing forces and adopt alternative binding modes or exit the active site altogether. This method allows us to interrogate the fidelity of the binding poses generated by the docking engine behind CovDock, Glide [35], and compare the stability of binding for different ligand/receptor pairings. Given the heterogeneity of the biasing forces (as the Gaussian biases are applied depending on stochastic ligand fluctuations), we ran ten replicates for each simulated system, for 10 nanoseconds each. Once completed, the hydrogen bond persistence (PersScore), time-dependent ligand RMSDs, and ligand final RMSD (PoseScore) were calculated using the Binding Pose Metadynamics tool. The PersScore term is calculated by measuring the fraction of frames in the last two nanoseconds of the simulations that have the same number of hydrogen bonds between the ligand and the receptor as the input structure, averaged over all of the replicates. Meanwhile, the PoseScore term is calculated by measuring the ligand's RMSD for the last frame of the simulation relative to its initial pose, also averaged across all replicates.

## Data analysis and visualization

Scores for ligands docked against mutant TEM structures were divided by the docking scores of the same ligand when docked against the TEM-1 structure, in order to obtain a measure of relative binding affinity. Occasionally, CovDock was not able to find favorable binding poses for certain compounds, due to insurmountable energy barriers created by steric clashes. For downstream calculations, the docking scores of these compounds against the tested receptor

were set equal to the docking score of the lowest-scoring compound for the same receptor. Mutants that led to occlusion of the binding site, such as Ala237Trp, were most likely to be rescored in this fashion.

Heat maps, scatter plots, and line plots were constructed using the Seaborn Python package [41] (Python version 3.7.15, Seaborn version 0.10.1). Hierarchical clustering of compounds by relative docking scores was performed using Seaborn's plotting engine and the Unweighted Pair-Group Method with Arithmetic Mean (UP-GMA) algorithm. [27]

Visualization of docked complexes and of interactions between compounds and residues of the binding site was performed using Schr̈odinger's Maestro (Schr̈odinger Release 2020–1: Maestro, Schr̈odinger, LLC, New York, NY, 2020). For non-bonded drug/target interactions, we used Maestro's default binary representation (present or absent) of non-bonded interactions (Schr̈odinger Release 2020–1: Maestro, Schr̈odinger, LLC, New York, NY, 2020). Ligand similarity was calculated in Schr̈odinger's Canvas [42] using circular extended connectivity fingerprints with search diameter 4 [33] as descriptors for leader-follower clustering, in which the ligand of interest was selected as the leader to identify compounds with structural homology (followers).

## Mutant generation

Starting with a circular construct containing wild-type TEM (pSkunk3-BLA, Addgene 61531), we used Agilent's QuikChange II Site-Directed Mutagenesis Kit along with a set of 38 oligonucleotide primers (S1 Table) to generate all standard amino acid substitutions at TEM's codon position 237. Then, we transformed separate aliquots of chemically-competent *E. coli* with each mutant construct. After confirming bacterial growth, we extracted plasmid DNA from individual colonies from each plate after overnight growth in liquid LB media. Finally, we confirmed the success of the mutagenesis by Sanger sequencing. Aliquots of bacteria expressing the desired mutant TEM from each isolated colony were stored as glycerol stocks (250 uL of bacteria-containing medium added to 250 uL of 50% a glycerol: ddH2O solution) at -80 C.

## Experimental measurements

Minimum-inhibitory concentration (MIC) assays were conducted using 96 well plates, with columns containing serial dilutions of each drug (ampicillin or cefixime), starting on the first column at a drug concentration of 256 $\mu$g/mL and ending on column 10 at a drug concentration of 0.5 $\mu$g/mL. Columns 11 and 12 were used as no-drug and no-bacteria controls, respectively. For each plate, each row was inoculated with a different mutant strain obtained above up to column 11. After overnight culture at 37$^{\circ}$C, we used a plate reader to assess turbidity (by measuring light absorption at 650 nm) and infer bacterial growth in each well and calculate the MIC for each mutant for both drugs. In order to quantify the effect of each mutation on fitness, we divided each mutant's MIC value upon treatment with each drug by the MIC value of the bacteria expressing wild-type TEM upon treatment with the same drug, resulting in a relative fitness value, which was used for validation of computational predictions. Each MIC assay was conducted in triplicates.

## Supporting information

**S1 Appendix. Choice of Docking Methodology.**
(PDF)

**S1 Fig. Summarized mechanism for the acylation reaction that inutilizes $\beta$-lactam antibiotics, catalyzed by TEM's Ser70.**
(PNG)

**S2 Fig. Local outcome of the Ala237Arg/Lys mutations in terms of binding site occlusion.** Serine 70, the catalytic residue, is represented in fuchsia.
(PNG)

**S3 Fig. Bi-dimensional representation of the scaffold shared by each of the three main drug classes tested in this study.**
(PNG)

**S1 Table. Sequences for the mutagenesis primers used to generate the 20 TEM Ala237X constructs.**
(PDF)

**S2 Table. List of all drugs used in the CovDock predictions.**
(PDF)

**S3 Table. CovDock scores (in kcal/mol) for the three compounds investigated in detail in this study against TEM-1 and TEM Ala237Arg/Lys.**
(PDF)

**S4 Table. Relative docking scores and fitness values for all tested mutants upon treatment with wither ampicillin or cefixime.** Mutants where no docking score was calculated due to extremely unfavorable poses had their scores set as" NAN".
(PDF)

**S1 Supporting Folder. Input coordinates (PDB format) for structures used in this study along with README file with further details.**
(ZIP)

## Acknowledgments

The authors thank Michael A. Stiffler for preliminary data that inspired this work. The authors thank Marty Ytreberg, Jagdish Patel, Dharmesh Patel and the many other members of the University of Idaho Center for Modeling Complex Interaction Molecular Modeling working group for numerous fruitful discussions, the authors also thank David Morgan for his help with setting up and running MIC assays.

## Author Contributions

**Conceptualization:** Gabriel Monteiro da Silva, Daniel M. Weinreich, Brenda M. Rubenstein.

**Data curation:** Brenda M. Rubenstein.

**Formal analysis:** Gabriel Monteiro da Silva.

**Funding acquisition:** Daniel M. Weinreich, Brenda M. Rubenstein.

**Investigation:** Gabriel Monteiro da Silva, Jordan Yang, Bunlong Leang, Jessie Huang.

**Methodology:** Gabriel Monteiro da Silva.

**Project administration:** Brenda M. Rubenstein.

**Resources:** Brenda M. Rubenstein.

**Supervision:** Daniel M. Weinreich, Brenda M. Rubenstein.

**Writing – original draft:** Gabriel Monteiro da Silva.

**Writing – review & editing:** Jordan Yang, Bunlong Leang, Daniel M. Weinreich, Brenda M. Rubenstein.

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
