## [Decision Letter · Decision Letter 0]

21 Apr 2022

Dear Mr Silva,

Thank you very much for submitting your manuscript "Covalent Docking and Molecular Dynamics Simulations Reveal the Specificity-Shifting Mutations Ala237Arg and Ala237Lys in TEM Beta-Lactamase" for consideration at PLOS Computational Biology. As with all papers reviewed by the journal, your manuscript was reviewed by members of the editorial board and by several independent reviewers. The reviewers appreciated the attention to an important topic. Based on the reviews, we are likely to accept this manuscript for publication, providing that you modify the manuscript according to the review recommendations.

Sincerely,

Bert L. de Groot

Associate Editor

PLOS Computational Biology

Nir Ben-Tal

Deputy Editor

PLOS Computational Biology

[LINK]

Reviewer's Responses to Questions

**Comments to the Authors:**

Reviewer #1: This manuscript explores how mutations give rise to new forms of beta-lactamase-mediated antibiotic resistance. It is a nice combination of computational methods with experimental tests of new predictions. The approach and insights could both help accelerate the development of new antibiotics to keep pace with the evolution of resistance.

- Briefly explain CovDock at the beginning of the results, or in the introduction. In particular, how is the covalent linkage enforced? Is there receptor flexibility? Is the ligand flexibility simulated on the fly, or pre-enumerated?

- From figure 1, it looks like Val, Leu, Gln, and Ile similar/more enhanced binding to cephalosporins. Is that not the case? If it is, what’s going on there? And why the focus on Lys/Arg?

- Are Ala237Lys/Arg mutations ever seen in the clinic? If not, why to the authors suppose that’s the case?

- Need error bars for all plots and results reported, so that the reader can judge their statistical significance.

- The correlation between the docking score and MIC is so low that I bet it is within error of zero. Either way, it may not be worth reporting, since the main conclusions are based on clear outliers (which are probably statistically significant), not a trend (which is probably absent).

- The results would be much stronger if the authors did the MIC measurements for the two new compounds, Cet and Car.

- The conclusions make the very interesting claim that Ala237 is highly conserved but that mutations can still increase fitness. Please show data to support the claim that the residue is conserved, compared to elsewhere in the protein or maybe even relative to other active site residues?

Reviewer #2: Monteiro da Silva and co-authors present a study on the effects of Ala237 mutations in TEM beta-lactamases. After performing covalent docking to screen 19 residue mutations at this position against a panel of 91 beta-lactam antibiotics, the authors further analyze the effects of A237R and A237K mutations with molecular dynamics simulations and Minimum Inhibitory Concentration (MIC) assays. Overall, this study provides convincing evidence of the opposite effects of these mutations against different classes of beta-lactam antibiotics and their underlying molecular mechanism. The manuscript is concise and clearly written, the work appears technically sound, and the discussion of the results is generally well balanced and measured with respect to the reliability of the different approaches and the overall evidence from current and past data.

Because of this, my opinion is that the manuscript is suitable for publication, although after a few minor revisions. My two main suggestions concern uncertainty estimation and coverage of past literature:

1) All results (figures 5, 6, and 7) concerning molecular dynamics simulations should report some measure of uncertainty/variation, such as mean and standard error. The qualitative difference observed seem to indicate that the different would also be statistically significant, but given the stochasticity of MD simulations, is importance to quantify the uncertainty associated with the observables studied. As such, Figure 5 should show the standard error of the mean or some confidence interval. Figures 6 and 7 should also show mean +/- uncertainty across the 10 repeats, rather than being taken from a single trajectory.

2) There have been several recent works on predicting drug resistance computationally via molecular modeling approaches. While the authors do cite some reviews in the introduction, I think it is appropriate to cite the relevant primary literature too in a more comprehensive appraisal of the field. This is a list of the studies I am aware of, but there are likely others:

Hauser et al. “Predicting resistance of clinical Abl mutations to targeted kinase inhibitors using alchemical free-energy calculations” Commun. Biol. 2018

Fowler et al. “Robust Prediction of Resistance to Trimethoprim in Staphylococcus aureus” Cell Chem. Biol. 2018

Aldeghi et al. “Accurate Estimation of Ligand Binding Affinity Changes upon Protein Mutation” ACS Cent. Sci. 2018

Aldeghi et al. “Predicting Kinase Inhibitor Resistance: Physics-Based and Data-Driven Approaches”, ACS Cent. Sci. 2019

Frey et al. “Predicting resistance mutations using protein design algorithms”, PNAS 2010

Brankin & Fowler “Predicting antibiotic resistance in complex protein targets using alchemical free energy methods”, ChemRxiv 2021

Sun et al “PremPLI: a machine learning model for predicting the effects of missense mutations on protein-ligand interactions” Commun. Biol. 2021

Yang et al. “SPLDExtraTrees: robust machine learning approach for predicting kinase inhibitor resistance” Brief. Bioinform. 2022

Other minor comments and suggestions:

- The validation of some of the docking results with more quantitative computational approaches, like free energy perturbation, could be a valuable and complementary addition to the study. I do not think it is strictly necessary for this work, however. This could be a consideration for future work.

- It looks like Figure S1 might be missing from the SI, as it does not match the description at page 5. Fig. S1 is about "Schrodinger’s CovDock workflow" but it is described in the text as "the side-chain oxygen of TEM's Ser70 residue binds to the /3-lactam's carbonyl via nucleophilic attack".

- There may be the opportunity to better highlight in Figures 1-3 which values correspond to "better vs worse" scores/fitness. E.g., in Figures 1 and 2, having the color bar be white when the relative score is equal to 1 (red when less than one, and blue when above one) would allow more immediately identify values close or slightly above/below 1; in Figure 3, perhaps one could label the relative fitness of 1 being the WT, with values above 1 meaning resistant mutations (worse drug efficacy), and below 1 sensitizing ones (better drug efficacy).

- In Figure 5, the PersScore and PoseScore are reported, but it is not clear what these are if one is not already familiar with them. It would be beneficial to add a brief explanation of these in the text or the figure caption directly.

- Somewhat related to the point above, from the Methods it is not clear how the PersScore and PoseScore are calculated. It would be good to add more details such that the reader will know what these scores are capturing more exactly.

**Have the authors made all data and (if applicable) computational code underlying the findings in their manuscript fully available?**

Reviewer #1: Yes

Reviewer #2: Yes

PLOS authors have the option to publish the peer review history of their article (what does this mean?). If published, this will include your full peer review and any attached files.

Reviewer #1: No

Reviewer #2: No

Figure Files:

Data Requirements:

Reproducibility:

References:

---

## [Editor Report · Decision Letter 1]

1 Jun 2022

Dear Mr Silva,

We are pleased to inform you that your manuscript 'Covalent Docking and Molecular Dynamics Simulations Reveal the Specificity-Shifting Mutations Ala237Arg and Ala237Lys in TEM Beta-Lactamase' has been provisionally accepted for publication in PLOS Computational Biology.

Best regards,

Bert L. de Groot

Associate Editor

PLOS Computational Biology

Nir Ben-Tal

Deputy Editor

PLOS Computational Biology

---

## [Editor Report · Acceptance letter]

22 Jun 2022

PCOMPBIOL-D-22-00270R1 

Covalent Docking and Molecular Dynamics Simulations Reveal the Specificity-Shifting Mutations Ala237Arg and Ala237Lys in TEM Beta-Lactamase

Dear Dr Silva,

I am pleased to inform you that your manuscript has been formally accepted for publication in PLOS Computational Biology. Your manuscript is now with our production department and you will be notified of the publication date in due course.

With kind regards,

Livia Horvath
